# The Third-Order Nonlinear Optical Properties of Sb$_2$S$_3$/RGO Nanocomposites

**Liushuang Li, Ye Yuan, Jiawen Wu, Baohua Zhu and Yuzong Gu ***

Physics Research Center for Two-Dimensional Optoelectronic Materials and Devices, School of Physics and Electronics, Henan University, Kaifeng 475004, China; lshuangl08@163.com (L.L.); yuanye_asuka@163.com (Y.Y.); 104753190687@henu.edu.cn (J.W.); bhzhu@henu.edu.cn (B.Z.)

\* Correspondence: yzgu@vip.henu.edu.cn

**Abstract:** Antimony sulfide/reduced graphene oxide (Sb$_2$S$_3$/RGO) nanocomposites were synthesized via a facile, one-step solvothermal method. XRD, SEM, FTIR, and Raman spectroscopy were used to characterize the uniform distribution of Sb$_2$S$_3$ nanoparticles on the surface of graphene through partial chemical bonds. The third-order nonlinear optical (NLO) properties of Sb$_2$S$_3$, RGO, and Sb$_2$S$_3$/RGO samples were investigated by using the Z-scan technique under Nd:YAG picosecond pulsed laser at 532 nm. The results showed that pure Sb$_2$S$_3$ particles exhibited two-photon absorption (TPA), while the Sb$_2$S$_3$/RGO composites switched to variable saturated absorption (SA) properties due to the addition of different concentrations of graphene. Moreover, the third-order nonlinear susceptibilities of the composites were also tunable with the concentration of the graphene. The third-order nonlinear susceptibility of the Sb$_2$S$_3$/RGO sample can achieve $8.63 \times 10^{-12}$ esu. The mechanism for these properties can be attributed to the change of the band gap and the formation of chemical bonds supplying channels for photo-induced charge transfer between Sb$_2$S$_3$ nanoparticles and the graphene. These tunable NLO properties of Sb$_2$S$_3$/RGO composites can be applicable to photonic devices such as Q-switches, mode-locking devices, and optical switches.

**Keywords:** Sb$_2$S$_3$/RGO composite; graphene; third-order nonlinear optical property; saturable absorption; susceptibility

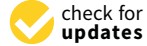



## 1. Introduction

Two-dimensional materials such as graphene, transition metal sulfides, and phosphorus have been widely studied in nonlinear optics. In order to explore novel NLO properties for suitable applications, these two-dimensional materials are often used to form composites with other materials. By changing the structure and composition, the NLO properties of the composite is expected to be altered or improved. Because graphene has many interesting electrical and optical properties due to its special bandgap structure [1,2], semiconductor–graphene composites have been extensively studied to obtain excellent electronic, magnetic, optical, catalytic, and mechanical properties [3–5]. In the semiconductor–graphene composites, graphene has been regarded as an excellent electron acceptor with the fastest known electron mobility of $1.5 \times 10^4$ m$^2$V$^{-1}$s$^{-1}$ [6]. In addition, monolayer graphene is easily dispersed in aqueous solution and polar solvent, which is conducive to the construction of graphene-based composites [7–9]. Some studies have shown that combining organic matter and transition metal sulfide with graphene can adjust its absorption properties to achieve improved optical limiting performances [10–12]. Some studies have combined transition metal sulfides, such as molybdenum disulfide and cadmium sulfide, with graphene to achieve the enhanced NLO properties [13,14].

Sulfide semiconductor nanomaterials have special third-order NLO properties [15–17]. Among them, Sb$_2$S$_3$ is one of the promising V-VI semiconductor materials that can be potentially applied in photoelectric sensors, near-infrared optical devices [18], optoelectronic

devices, and lithium-ion batteries [19]. The third-order NLO properties of $Bi_2S_3$ and $Sb_2S_3$ monomers and $Bi_2S_3$/RGO composites have been studied [20,21]. The application prospect of $Sb_2S_3$/RGO in photo degradation activity and a high-performance sodium ion battery has been explored. However, the third-order NLO properties of $Sb_2S_3$/RGO composites have been rarely discussed.

The main motivation of this work was to explore the third-order nonlinear optical properties of $Sb_2S_3$/RGO by successfully combining $Sb_2S_3$ nanoparticles with graphene and to enhance the third-order nonlinear properties of the composite by changing the concentration of GO. In this work, $Sb_2S_3$/RGO composites were synthesized via a facile, one-step solvothermal method. The third-order NLO properties were tested by using Z-scan technique with a 30-ps laser at 532 nm. The mechanism for the NLO properties was analyzed. The tunable saturable absorption and positive nonlinear refraction properties of $Sb_2S_3$/RGO composites were obtained, which can be applicable to the Q-switch, mode-locking, and all-optical switch.

## 2. Materials and Methods

### 2.1. Sample Preparation

2.1.1. Synthesis of GO

GO was prepared by a modified Hummers method [22,23]. First, a small amount of concentrated sulfuric acid ($H_2SO_4$) and concentrated phosphoric acid ($H_3PO_4$) were mixed into a three-necked flask. The ground mixture of powdered graphite and potassium permanganate was then slowly added to the three-necked flask. Then, the three-necked flask heated to 50 °C and stirred for 24 h. After the reaction was completed, an appropriate amount of diluted hydrogen peroxide solution was added to the reactants. Then, the reaction products were treated with ultrasound and washed with hydrochloric acid and deionized water once and four times, respectively. Finally, the supernatant was removed by centrifugal process and the product was freeze-dried in a vacuum.

2.1.2. Synthesis of $Sb_2S_3$/RGO Composites

A certain mass of GO was dispersed into the corresponding ethylene glycol (EG) and then treated with ultrasound for 2 h to obtain a clear solution of GO-EG at the concentration of 0.1 mg/mL. Then, 0.114 g of antimony chloride ($SbCl_3$), 0.15 g of polyvinylpyrrolidone (PVP), and 0.114 g of thiourea were added to the GO-EG solution to form a reaction mixture. After the mixed solution was stirred for about 30 min, the reaction mixture was transferred to a stainless steel autoclave with a PTFE lining, heated at 100 °C for 12 h, and cooled to ambient temperature naturally. The resulting precipitate was collected by centrifugation, washed with ethanol and water several times, and vacuum-dried overnight for characterization. The final product was a $Sb_2S_3$/RGO composite prepared at 0.1 mg/mL GO concentration, expressed as G2. In the control experiment, a series of $Sb_2S_3$/RGO composites were synthesized by fixing the amount of $Sb^{3+}$ and thiourea and changing the GO concentration to 0.05 mg/mL, 0.5 mg/mL, and 1 mg/mL. The products were expressed as G1, G3, and G4, respectively. For comparison, bare $Sb_2S_3$ was synthesized through the same steps without GO, and graphene was synthesized without using $SbCl_3$ and thiourea. Figure 1 shows schematically the possible formation mechanism of $Sb_2S_3$ and the $Sb_2S_3$/RGO materials.

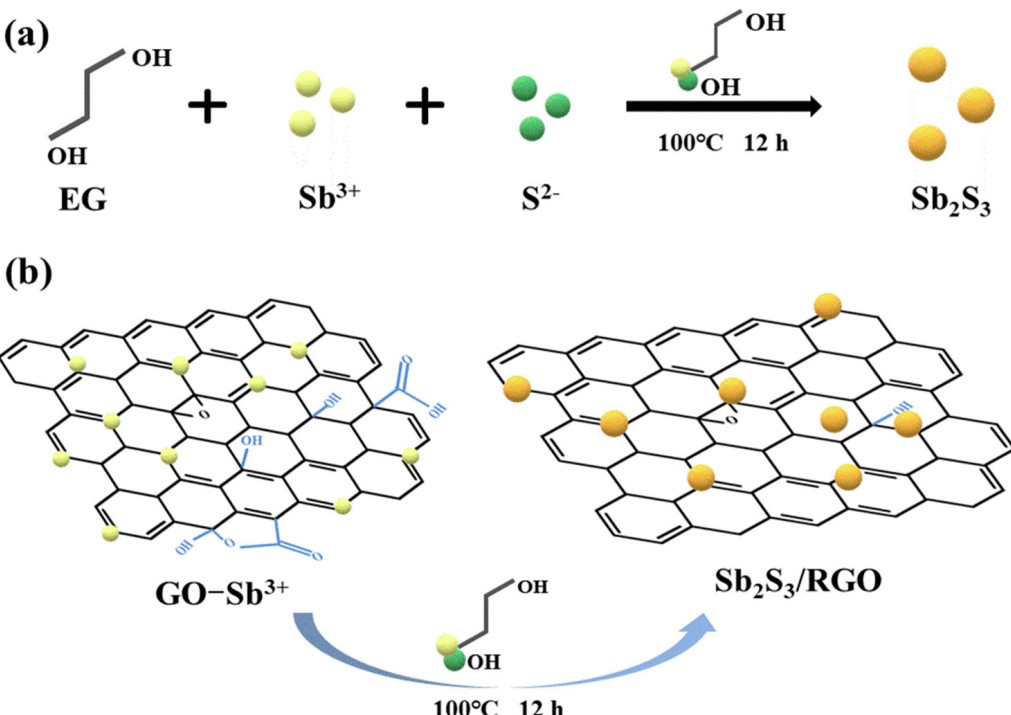

**Figure 1.** Schematic illustration of the formation of Sb$_2$S$_3$ NPs (**a**) and Sb$_2$S$_3$/RGO (**b**).

*2.2. Sample Characterization*

The morphology and structure of the samples were characterized through field emission scanning electron microscope (SEM, Carl Zeiss Inc., Oberkochen, Baden-Württemberg, Germany), transmission electron microscope (TEM, JEOL JEM-2100 working at 200 kV, JEOL Ltd. Inc., Akishima, Tokyo, Japan), and X-ray diffraction (XRD, Bruker D8 Advance, Bruker Inc., Karlsruhe, Badensko-Wuertembersko, Germany). The functional groups of samples were identified by Fourier transform infrared spectroscopy (FTIR, VERTEX 70v Bruck Optics, Germany). Raman spectra were obtained by a Raman spectrometer. The UV-Vis spectra were measured on a PerkinElmer Lambda 35 ultraviolet-visible spectrometer (Agilent Inc., Sacramento, CA, USA). The laser source used for the NLO measurement was an Nd:YAG laser system (EKSPLA, PL2251) with a wavelength of 532 nm, a pulse width of 30 ps, and a pulse repetition of 10 Hz.

### 3. Results

*3.1. Structural and Morphology Characterization*

The XRD patterns of Sb$_2$S$_3$ and Sb$_2$S$_3$/RGO powders are shown in Figure 2. It can be seen that the Sb$_2$S$_3$/RGO composites showed weak diffraction peaks, in the range of $15° \sim 60°$ [24], and no clear diffraction of other impurities was detected, indicating that the Sb$_2$S$_3$/RGO powders had good composite properties. The morphology and phase of Sb$_2$S$_3$ were affected by solvothermal temperature. When the solvothermal reaction was carried out at 100 °C, the phase of the product was amorphous (Figure 2), indicating that the temperature was too low to crystallize Sb$_2$S$_3$ [25]. The lack of diffraction peaks in the sample revealed the amorphous nature. In addition, there was no grapheme diffraction peak, possibly because the modification of Sb$_2$S$_3$ on the graphene sheet destroyed the orderly stacking of the graphene sheet and the uneven spacing of the graphene layers led to the disappearance of diffraction belonging to graphene.

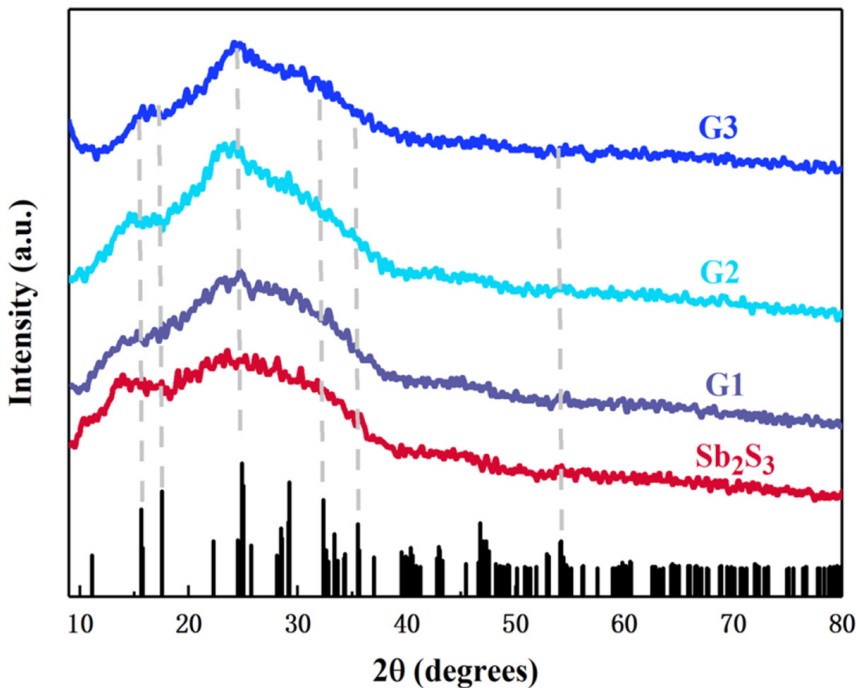

**Figure 2.** XRD patterns of the $Sb_2S_3$ and $Sb_2S_3$/RGO (G1–G3).

The morphology and structural characteristics of the $Sb_2S_3$/RGO composites synthesized with different concentrations of GO were characterized by scanning electron microscopy. Figure 3a shows that the $Sb_2S_3$ monomers were spherical structures with a diameter of about 1 μm. However, when they were growing on graphene, as shown in Figure 3b–d, the size and distribution of the $Sb_2S_3$ particles were affected by the layered graphene. Due to the anisotropy of the $Sb_2S_3$ nanoparticles, some of them were uniformly distributed on the graphene while others were clustered together with a rough surface and a size of about 120 nm (Figure 3b). A further increase in the GO concentration resulted in insufficient coverage of the $Sb_2S_3$ nanoparticles on the graphene sheet, and the $Sb_2S_3$ nanoparticles sparsely covered the graphene sheet with a slight decrease in size (Figure 3c,d). Based on the above results, the reaction of antimony chloride with thiourea may have produced unstable antimony compounds, while the $Sb_2S_3$ NPs produced amorphous particles on the surface of the graphene through the heterogeneous nucleation process [26]. The antimony compounds were dispersed into the $Sb_2S_3$ nanoparticles by the solvothermal method. Coordination of COOH and OH groups with Sb (III) promoted good distribution of the $Sb_2S_3$ nanoparticles on graphene sheets [27]. Therefore, the combination of RGO and $Sb_2S_3$ not only made $Sb_2S_3$ adhere to the graphene layer but also improved the conductivity of $Sb_2S_3$.

In order to further investigate the structural characteristics of $Sb_2S_3$/RGO, the prepared materials were observed by transmission electron microscopy (TEM) and their structures were consistent with the results of the SEM analysis. Figure 3e,f shows the TEM images of the $Sb_2S_3$/RGO composites; it can be seen that the $Sb_2S_3$ monomer synthesized by PVP was spherical and dispersed nanoparticles were attached to the corrugated layered structure of the graphene, while some of them were clustered on the edge of graphene [28].The $Sb_2S_3$ NPs were deposited on the surface with oxygen-containing groups in GO reactants as anchor sites, indicating that the $Sb_2S_3$ NPs were successfully supported on the surface of RGO, which prevented aggregation of the $Sb_2S_3$ NPs [27]. The sufficient connection between the $Sb_2S_3$ nanoparticles and graphene maximized the efficiency of electron transfer due to the sufficient density and good dispersion of the relatively large $Sb_2S_3$ NPs on the graphene sheet.

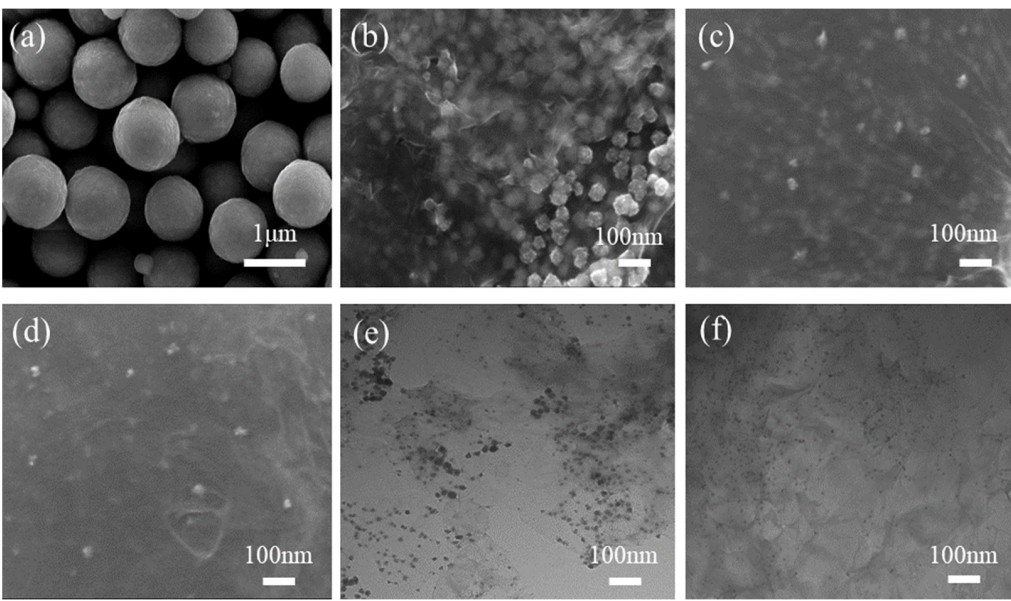

**Figure 3.** SEM images of $Sb_2S_3$ (**a**) and the $Sb_2S_3$/RGO composites G2 (**b**), G3 (**c**), G4 (**d**); TEM images of composites G2 (**e**) and G3 (**f**).

To investigate the characteristic vibration modes of the prepared samples, Raman spectroscopy was carried out with the excitation of 532 nm. Figure 4 shows the Raman spectra of pure GO, $Sb_2S_3$, and $Sb_2S_3$/RGO with different concentrations of GO. Raman spectra of pure $Sb_2S_3$ and $Sb_2S_3$/RGO showed that the diffraction peaks were about 300 $cm^{-1}$. Both GO and $Sb_2S_3$/RGO showed two main peaks at 1349 $cm^{-1}$ and 1599 $cm^{-1}$, which were correlated with the disorder (D) band and graphite (G) band of carbon-based materials, respectively, and the D band was related to the defect state of graphene. $I_D/I_G$ represents the carbon atom ratio of $SP^2/SP^3$, indicating that GO had a large number of oxygen-containing functional groups [29]. Compared with GO ($I_D/I_G = 0.95$), $Sb_2S_3$/RGO ($I_D/I_G = 1.01$) showed an increased D-G strength ratio, indicating there were more disordered carbons and plenty of surface defects [30]. These defects affected the charge distribution and properties of the composites. This was caused by the decrease in the average size of $SP^2$ domains and the increase in the number of domains. The change in $I_D/I_G$ intensity usually indicates the reduction of graphene oxide to graphene [31] and can also be widely used to evaluate the structural defects of graphene materials [32].

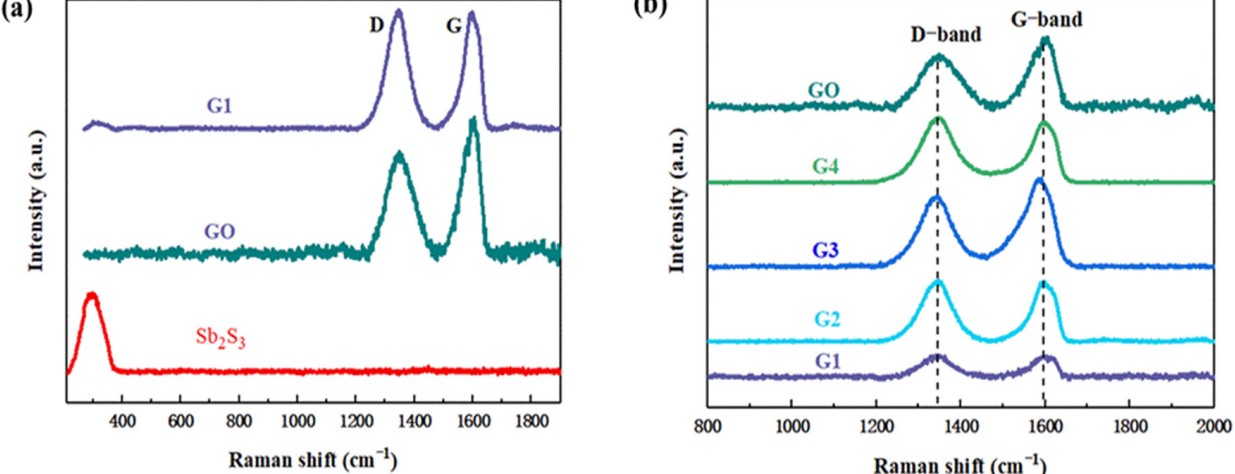

**Figure 4.** Raman patterns of GO, $Sb_2S_3$ and G1 (**a**); Raman patterns of GO and composites G1–G4 (**b**).

Figure 5a shows the infrared spectra of GO, $Sb_2S_3$, and $Sb_2S_3$/RGO composites prepared with different GO concentrations. For GO, the broad and strong peak at 3379 cm$^{-1}$ was attributed to the stretching vibration of the -OH group, the peak at 1701 cm$^{-1}$ represents the C=O vibration of -COOH located at the edge of the GO sheet, and the peak at 1638 cm$^{-1}$ was attributed to O-H bending vibration, epoxy group, and aromatic C=C skeleton tensile vibration [33]. The peaks at 1207 cm$^{-1}$ and 1068 cm$^{-1}$ were attributed to the C-O-C or C-O-H stretching and C-O stretching of the carboxyl group in GO. The FT-IR pattern of G4 (see Figure 5b) confirmed the formation of Sb-S bonds due to the appearance of absorption peaks at 538 cm$^{-1}$, 634 cm$^{-1}$, and 721 cm$^{-1}$. Additionally, a new peak appeared at 994 cm$^{-1}$, which may be attributed to C–S stretching vibrations and also demonstrated the strong bonding of GO and $Sb_2S_3$.

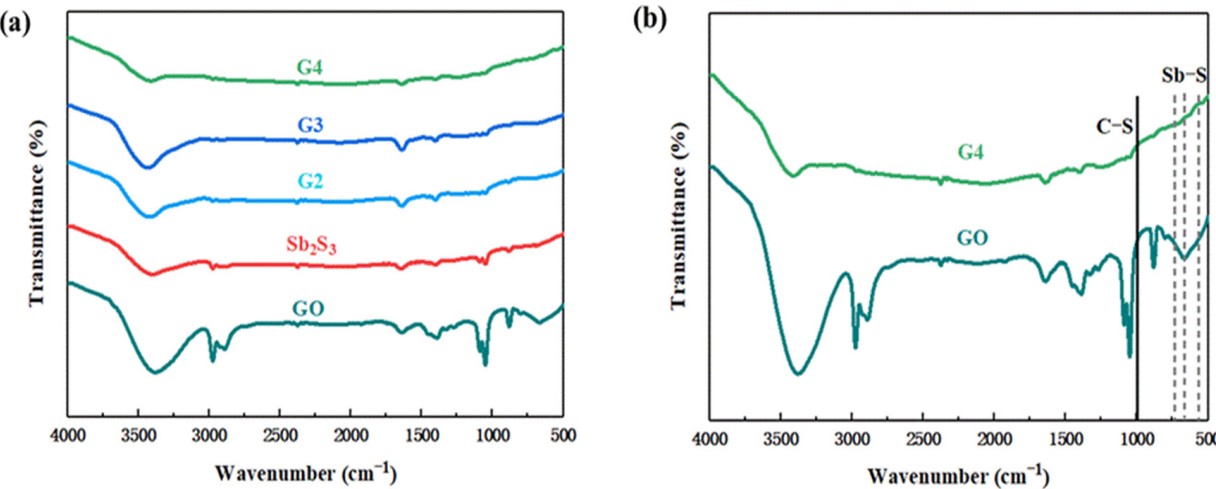

**Figure 5.** FT−IR spectra of GO, $Sb_2S_3$, and the $Sb_2S_3$/RGO composites G2–G4 (**a**); FT-IR spectra of GO and G4 (**b**).

The representative peaks of the carboxyl and hydroxyl groups of GO appeared in the composite materials, indicating that the $Sb_2S_3$/RGO composites were successfully synthesized by a solvothermal reaction. The strong covalent bonds and weak van der Waals forces between Sb and S atoms justified the formation of $Sb_2S_3$ nanoparticles, which, in turn, succeeded in firmly bonding to graphene due to the strong chemical interaction between carbon and sulfur. However, the peaks of oxygen-containing groups in the composites were significantly weakened, which indicated that the solvothermal reaction in EG removed most of the carboxyl and hydroxyl groups [34]. According to the infrared spectrum of G2 and G3 in the figure, it can be seen that the C=C framework vibration of the graphene sheet (1638 cm$^{-1}$) was prominent in the composite material, which also confirmed the recovery of the aromatic SP$^2$ hybridized carbon skeleton of graphene [35]. These results indicated that the solvothermal reaction can effectively reduce graphene oxide to graphene.

### 3.2. Linear Optical Properties

To preliminarily understand the nonlinear optical absorption (NOA) mechanism, the linear optical properties of the prepared samples $Sb_2S_3$ and $Sb_2S_3$/RGO composites were characterized by UV-Vis diffuse reflectance spectroscopy (UV-Vis DRS). As shown in Figure 6, the $Sb_2S_3$ NPs had some absorption at the excitation wavelength of 610 nm. However, they had more absorption at 240 nm, indicating the probability of excited states' absorption in two steps or genuine two-photon absorption [36]. The $Sb_2S_3$/RGO had a wide shoulder peak after 270 nm, and the absorption peak had a red shift. It was reported that $Sb_2S_3$ is a direct band gap semiconductor material and has a high absorption rate [37]. The relation between the light absorption coefficient $\alpha$ and the energy of incident light photon $h\nu$ near the absorption edge was as follows [38]: $\alpha h\nu = (h\nu - E_g)^{1/2}$ ($E_g$ is the band gap width). Therefore, the relation between $(\alpha h\nu)^2$ and $(h\nu)$ near the absorption edge

was linear and the band gap $E_g$ could be calculated. According to Beer's law, absorbance A = abc, b is the sample thickness in cm, c is the sample concentration in g/L, and the proportional coefficient alpha is called the absorption coefficient. A can also be defined as A = −lgT, where T is the transmission; so, the definition of alpha can also be expressed as T = exp(−alpha *c*b) in $L \cdot g^{-1} \cdot cm^{-1}$. In addition, the value of linear absorption coefficient can also be obtained: $\alpha$ ($Sb_2S_3$) = 0.77 $L \cdot g^{-1} \cdot cm^{-1}$, $\alpha$ (G1) = 0.24 $L \cdot g^{-1} \cdot cm^{-1}$, $\alpha$ (G2) = 0.21 $L \cdot g^{-1} \cdot cm^{-1}$, $\alpha$ (G3) = 0.70 $L \cdot g^{-1} \cdot cm^{-1}$, and $\alpha$ (G4) = 0.54 $L \cdot g^{-1} \cdot cm^{-1}$. The absorption spectra of the samples indicated the presence of different local levels in the forbidden gap, which may have originated from amorphous defects in the antimony sulfide nanostructure, such as vacancies and surface defects. The presence of defects may somewhat affect the band gap structure [39]. The $Sb_2S_3$/RGO composites had a reduced band gap compared to $Sb_2S_3$, which provided the evidence of a connection between antimony sulfide and graphene. The difference in the absorption curves of the composites and the $Sb_2S_3$ NPs can be explained by the presence of RGO and the energy interaction between the two materials [40].

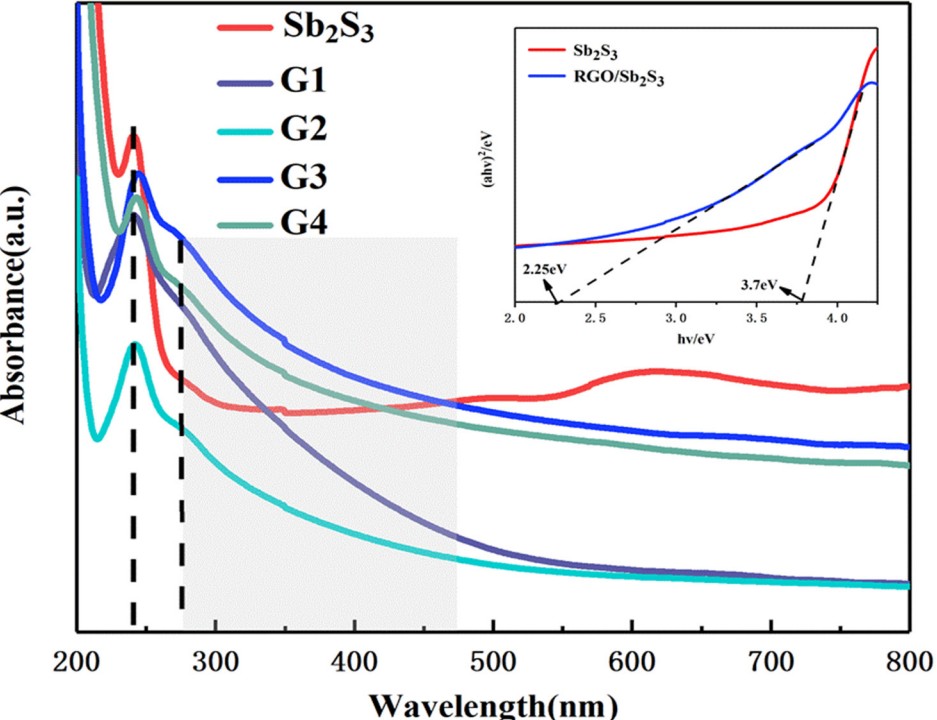

**Figure 6.** UV–Vis absorption spectra of $Sb_2S_3$ and the $Sb_2S_3$/RGO composites.

### 3.3. Nonlinear Optical Properties

The Z-scan curves in Figure 7 show the NLO characteristics of the sample. The NLO properties of the $Sb_2S_3$ and $Sb_2S_3$/RGO composites were studied by using Z-scan technique with a laser at wavelength of 532 nm, pulse width of 30 ps, and repetition rate of 10 Hz. The interval between two pulses was about 0.1 s, which was much longer than the pulse width. Therefore, a thermal effect can be ignored. Actually, the laser used had the same effect as a single-shot mode laser. The samples were dissolved in ethanol solvent to prepare solutions with a concentration of 0.1 mg/mL. The optical path length (cuvette) was 1 mm. The actual thickness of the sample was about 0.4 mm, including the total thickness of the cuvette minus its double walls. Therefore, the sample thickness was considered to be much less than the Rayleigh range; thus, the thin sample approximation was satisfied. It was found that the NLO signal of ethanol solvent at 532 nm was very small and could be neglected [41]. Therefore, the nonlinearity of the solvent could be ruled out and the obtained Z-scan curves directly showed the nonlinear response of graphene [42], $Sb_2S_3$, and $Sb_2S_3$/RGO.

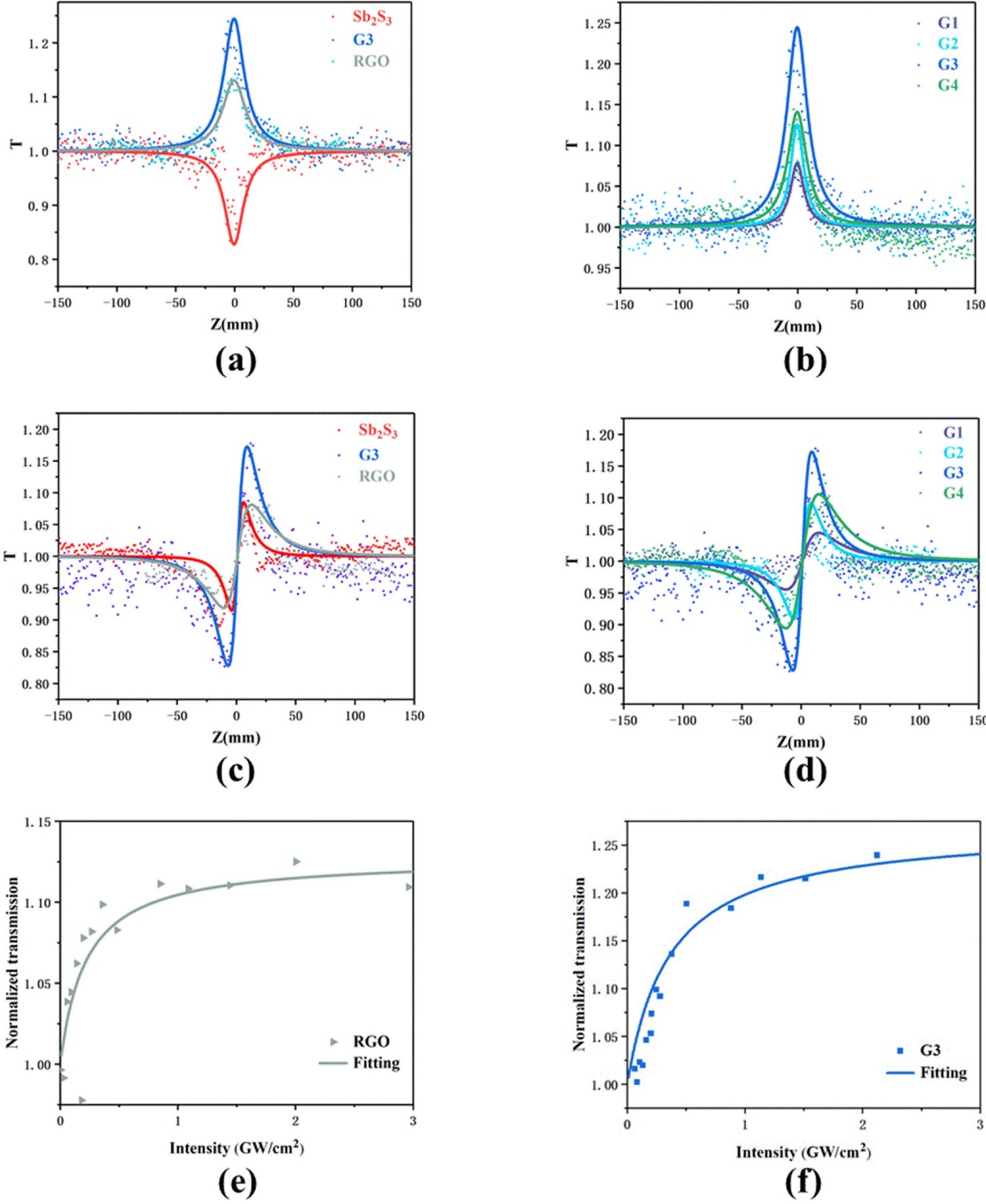

**Figure 7.** (**a**) Open-aperture Z-scan curves of $Sb_2S_3$, RGO, and the $Sb_2S_3$/RGO composite G3; (**b**) open-aperture Z-scan curves of composites G1–G4. (**c**) Closed-aperture/open-aperture Z-scan curves of $Sb_2S_3$, RGO, and the $Sb_2S_3$/RGO composite G3; (**d**) closed-aperture/open-aperture Z-scan curves of composites G1–G4. (**e**,**f**) Normalized transmission versus input intensity of RGO and G3.

The curves in Figure 7a are the Z-scan curves of $Sb_2S_3$, G3 ($Sb_2S_3$/RGO-0.5), and RGO, reflecting the variation of nonlinear absorption characteristics. The curves of RGO and $Sb_2S_3$/RGO appear as symmetrical peaks at the focus, indicating the dominant position of saturation absorption (SA) in the NLO absorption mechanism; the saturated absorption

of G3 shows an increasing trend compared with that of G1–G2. Generally, SA happens in the resonant region where the photon energy is larger than or equal to the band gap of the materials due to the electrons' excitation from valence band to conduction band and then the Pauli-blocking induced bleaching effect [43]. In the state of sufficiently large light intensity, electrons in the valence band can jump to the conduction band through single-photon absorption. The photon bands of the valence band and conduction band were completely occupied by electrons and holes; therefore, further light absorption was blocked and SA occurred [44]. The $Sb_2S_3$ NPs exhibited two-photon absorption (TPA), which can be seen in Figure 7a, due to the fact that the band gap of $Sb_2S_3$ NPs was greater than the excitation photon energy. Figure 7b shows the OA Z-scan curves (G1–G4) of $Sb_2S_3/RGO$ corresponding to different GO concentrations. From G1–G4, as the graphene increased, the peaks first increased and then decreased. This was due to the fact that when the graphene content was increased to a certain amount (G4), the lower $Sb_2S_3$ content in these composite materials could only generate a low level of photo-excited electron-hole pairs under irradiation. It can be seen from Figure 7c,d that all CA/OA Z-scan curves produced typical valley–peak trajectories, which indicated that all the samples had positive nonlinear refraction and self-focusing characteristics.

The theoretical fitting curves, shown as a solid line in Figure 7, and the third-order nonlinear parameters were obtained using the Z-scan theory. The normalized Z-scan curves of OA were well fitted by the following equation [45], which is used to describe a third-order NLO absorptive process:

$$T(z) = \sum_{m=0}^{\infty} \frac{[-q_0(z)]^m}{(1+m)^{3/2}} \tag{1}$$

where $q_0(z) = (\beta I_0 L_{eff})/(1 + z^2/z_0^2)$ and $\beta$ is the nonlinear absorption coefficient. The $z_0 = \pi \omega_0^2 / \lambda$ is the length of Rayleigh diffraction, which can be calculated to be 0.66 mm. $I_0$ is the laser intensity at the focus (1.69 $GW/cm^2$), $L_{eff} = [1 - \exp(-\alpha L)]/\alpha$ is the effective length of the sample, L is the actual thickness of the sample, and $\alpha$ is the linear absorption coefficient of the sample, which can be obtained from the UV-Vis absorption spectra. When $q_0 < 1$, the normalized transmittance formula is approximated to the following formula:

$$T(z) \approx 1 - \frac{\beta I_0 L_{eff}}{2\sqrt{2(1 + z^2/z_0^2)}} \tag{2}$$

According to the above equation, $\beta = 2^{3/2}(1 - T_{z=0})(1 + z^2/z_0^2)/I_0 L_{eff}$ can be obtained, and the nonlinear absorption coefficient of the sample can be calculated by this formula. The imaginary part of the third-order nonlinear susceptibility is gained by $Im\chi^{(3)} = cn_0^2 \lambda \beta / 480\pi^3$ and the normalized CA/OA Z-scan transmittance is calculated as [46]:

$$T(z) = 1 - \frac{4x\Phi_0}{(x^2 + 9)(x^2 + 1)} \tag{3}$$

where $x = z/z_0$; so, the third-order nonlinear refraction index is $n_2(esu) = (cn_0/40\pi)\gamma(m^2/W)$. The $\gamma$ is the nonlinear refractive index coefficient in $m^2/W$ and $\Delta T_{p-v}$ is the peak-to-valley difference. The relation between $\gamma$ and $\Delta T_{p-v}$ can be expressed by $\gamma = \lambda \alpha \Delta T_{p-v}/[0.812\pi I_0 (1 - S)^{0.25}(1 - e^{-\alpha L})]$. S is the linear transmittance of the aperture. The $\Delta \Phi_0$ is the onaxis phase shift at the center focus, which can be expressed by the formula $\Delta \Phi_0 = k\Delta n_0 L_{eff} = k\gamma I_0 L_{eff}$ containing $\gamma$. The real part of the third-order nonlinear susceptibility is $Re\chi^{(3)} = n_0 n_2/3\pi$ and the third-order nonlinear susceptibility is given by $\chi^{(3)} = \left[\left(Re\chi^{(3)}\right)^2 + \left(Im\chi^{(3)}\right)^2\right]^{1/2}$.

The data listed in Table 1 are the third-order nonlinear parameters of the samples (RGO, $Sb_2S_3$, $Sb_2S_3/RGO$) calculated according to the above formula. From $Sb_2S_3$ to $Sb_2S_3/RGO$,

the imaginary part of third-order nonlinear optical susceptibility $Im\chi^{(3)}$ changes from negative to positive and the $Im\chi^{(3)}$ values of G1–G3 increase, indicating that the nonlinear absorption can be regulated. It also shows the dominant role of saturated absorption in the recombination process [47]. In addition, the real $Re\chi^{(3)}$ associated with the Kerr effect was also regulated and enhanced, and its maximum value was $8.03 \times 10^{-12}$ esu, which appeared in G3. The third-order nonlinear susceptibility $\chi^{(3)}$, determined by $Im\chi^{(3)}$ and $Re\chi^{(3)}$, is thus adjusted and enhanced to a maximum of $8.63 \times 10^{-12}$ esu. The $Sb_2S_3$ NPs were evenly distributed on the graphene as the graphene increased. Even though the distribution of NPs was random, there existed a Fabry–Perot-like localized resonance due to small-scale ordering within the architecture, which resulted in the local enhancement of the field [48]. This local field enhancement led to an increase in NLO interactions and, hence, higher values for $n_2$ and $\beta$.

**Table 1.** The NLO parameters of the samples.

| Sample | $\beta$ ($10^{-11}$ m/w) | $\gamma$ ($10^{-18}$ m²/w) | $Im\chi^{(3)}$ ($10^{-12}$ esu) | $Re\chi^{(3)}$ ($10^{-12}$ esu) | $\chi^{(3)}$ ($10^{-12}$ esu) |
|---|---|---|---|---|---|
| RGO | −2.17 | 0.03 | −1.36 | 0.04 | 1.36 |
| $Sb_2S_3$ | 2.60 | 2.42 | 1.96 | 4.31 | 4.73 |
| G1 | −1.32 | 0.39 | −0.99 | 0.69 | 1.21 |
| G2 | −2.11 | 0.68 | −1.59 | 1.21 | 2.00 |
| G3 | −4.22 | 4.51 | −3.18 | 8.03 | 8.63 |
| G4 | −2.42 | 2.07 | −1.82 | 3.69 | 4.11 |

Figure 7e,f shows the relationship between normalized transmittance and saturation intensity of samples, which further shows that the composite and RGO exhibited the same absorption characteristics, corresponding to Figure 7a. We can fit the relevant data by the following formula:

$$T = 1 - \frac{\alpha_s}{1 + \dfrac{I}{I_{sat}}} - \alpha_{ns} \tag{4}$$

where $\alpha_{ns}$ is the non-saturated component, $\alpha_s$ is the modulation depth, and $I_{sat}$ is the saturated intensity. We can obtain the $\alpha_S$ of 6.2%, 11.2%, 21.0%, and 10.7% for the G1, G2, G3, and G4 corresponding to the $I_{sat}$ of 0.56 GW/cm², 0.47 GW/cm², 0.36 GW/cm², and 0.41 GW/cm². According to the relevant nonlinear parameters in Tables 1 and 2, it can be concluded that G3 had the highest nonlinear absorption coefficient $\beta = -4.22 \times 10^{-11}$ m/w and the lowest saturation strength $I_{sat} = 0.36$ GW/cm². The saturated intensity of the sample (G1–G3) decreased, which corresponded to the change in the nonlinear absorption characteristics shown in Figure 7b. Comparing the nonlinear parameters of different materials, we found $Sb_2S_3$/RGO had strong third-order nonlinear optical properties. This shows that $Sb_2S_3$/RGO composites have potential applications in mode locking and pulse compression.

**Table 2.** The relevant nonlinear parameters of different materials.

| Sample | $\lambda$ (nm) | $\beta$ ($10^{-11}$ m/w) | $\gamma$ ($10^{-18}$ m²/w) | $I_s$ (GW/cm²) | Reference |
|---|---|---|---|---|---|
| $Sb_2S_3$ | 532 | 2.60 | 2.42 | − | This work |
| $Sb_2S_3$/RGO (G3) | 532 | −4.22 | 4.51 | 0.36 | This work |
| $MoSe_2$ | 1064 | −2.05 | − | 0.71 | [49] |
| G-CuO | 1030 | 1.37 | 0.48 | − | [50] |

It can be seen from the data that the third-order nonlinearity of $Sb_2S_3$/RGO was regulated. It is well known that the NLO process is controlled by the nonlinear susceptibility $\chi^{(3)}$ of NLO materials. The higher the $\chi^{(3)}$ value is, the better the NLO performance. To further understand the variation of nonlinear optical absorption in the composite structure, we analyzed its possible photo-induced charge carrier transfer behavior [51], as shown in Figure 8. Due to the existence of energy band differences between $Sb_2S_3$ and RGO, a donor–acceptor electronic structure was formed in the complex. $Sb_2S_3$ can be regarded as an electron donor, and RGO as an electron acceptor, so that electrons cannot only transit within each of them, but also transfer between them. Additionally, the photo-generated electrons in $Sb_2S_3$ NPs would immigrate to RGO and further be trapped by the surface defects in RGO [52], which may minimize the possibility of recombination of photo-generated carriers. The excited electrons are transferred from the conduction band of $Sb_2S_3$ to the graphene and then to the valence band of $Sb_2S_3$. The progress may interrupt the carrier relaxation in RGO and favor SA.

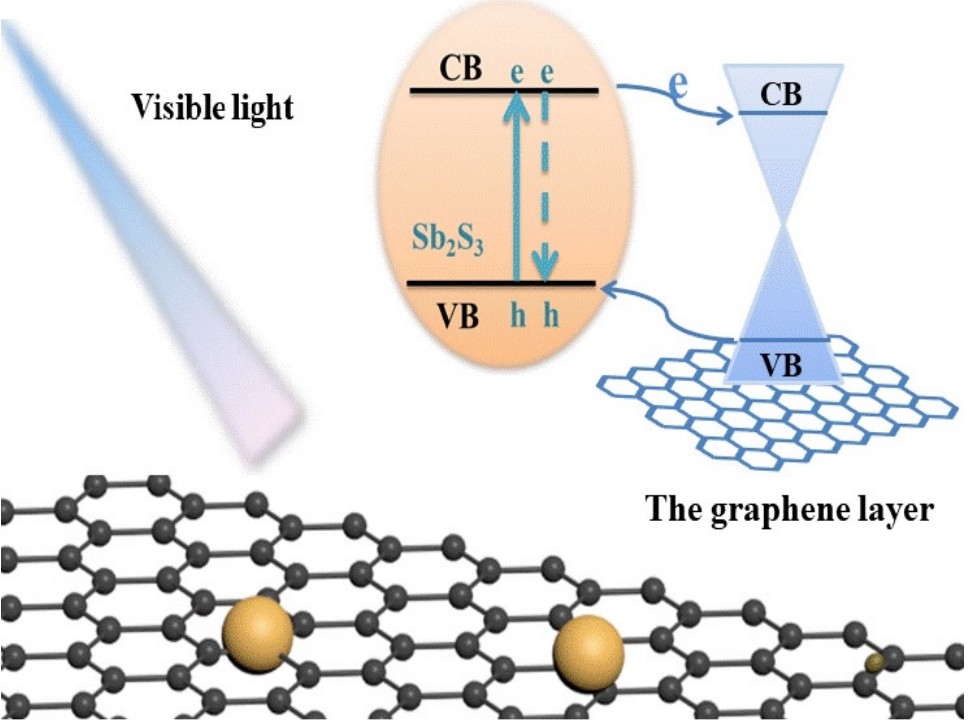

**Figure 8.** Charge-transfer mechanism of the $Sb_2S_3$/RGO composites.

### 4. Conclusions

In conclusion, the $Sb_2S_3$/RGO composites were successfully synthesized by the facile, one-step solvothermal method and the NLO properties of all samples were studied at 532 nm by the Z-scan technique. It was found that the addition of GO transformed the two-photon absorption of $Sb_2S_3$ into adjustable saturation absorption of $Sb_2S_3$/RGO composites, which was attributed to the change of band gap. The tunable positive nonlinear refraction properties and the enhanced nonlinear susceptibility $\chi^{(3)}$ of $Sb_2S_3$/RGO composites were obtained, which were larger than $Sb_2S_3$ and more than six times that of RGO. The mechanism of its nonlinear optical properties was believed to be that the effective charge and energy transfer between $Sb_2S_3$ NPs and RGO enhance the free carrier absorption and nonlinear refraction process. It was directly revealed that the $Sb_2S_3$/RGO composites have tunable nonlinear susceptibility $\chi^{(3)}$ with different GO concentrations. The results of these tunable third-order NLO properties of the $Sb_2S_3$/RGO composites would provide the basis for the application in photonic devices.

**Author Contributions:** Conceptualization, L.L. and Y.Y.; methodology, L.L.; validation, J.W. and Y.Y.; formal analysis, L.L.; investigation, L.L.; resources, Y.G.; writing—original draft preparation, L.L.; writing—review and editing, Y.G.; supervision, B.Z.; project administration, Y.G.; funding acquisition, Y.G. and B.Z. All authors have read and agreed to the published version of the manuscript.

**Funding:** This research was funded by the National Natural Science Foundation of China (61875053, 61404045, U1404624) and Excellent Youth Project of Henan Province of China (202300410047).

**Institutional Review Board Statement:** Not applicable.

**Informed Consent Statement:** Not applicable.

**Data Availability Statement:** The data presented in this paper are available from the authors upon reasonable request.

**Conflicts of Interest:** The authors declare no conflict of interest.

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
