# Peer review of "The Third-Order Nonlinear Optical Properties of Sb2S3/RGO Nanocomposites"

_photonics, doi:10.3390/photonics9040213_

Round 1
Reviewer 1 Report
Review
Manuscript ID: photonics-1615645
Title: The third-order nonlinear optical properties of Sb2S3/RGO nanocomposites.
Authors: Liushuang Li, Ye Yuan, Jiawen Wu, Yuzong Gu, Baohua Zhu
Recent Advances in Nonlinear Optics and Nonlinear Optical Materials.
I read this article with great interest. It seems well written, and the results are relevant. There would be some improvements to make it acceptable for publication. Concerning the part on specimen preparation, synthesis and linear characterization I am not a specialist, and I can't judge it. But the part dealing with Z scan experimentation and non-linear characterizations is an element of my expertise.
Let's start with figure 6 where the absorbance is not graduated there is no value corresponding to the y-axis it would give more meaning to the article to put the absorbance absolute unit and not arbitrary one.
It is necessary to give the values of the linear absorption in the table for each specimen, there would probably and certainly be a correlation between the NL values found and these values.
Thank you also for pointing out the intensities used and for discussing a few words about whether there was a possible destruction induced by the high intensities and how can we be sure that the irradiated material has not been modified.
I will not quote all references that are not adequate but when appealing to equations typically related to the basic Z-scan article, reference 38 is not the first one you think of, it should be replaced by [Sheik-Bahae, M., Said, A. A., Wei, T. H., Hagan, D. J., & Van Stryland, E. W. (1990). “Sensitive measurement of optical nonlinearities using a single beam”. IEEE journal of quantum electronics, 26(4), 760-769.]…
In addition, the article would benefit from calling on other works and comparisons with some studies done on other nanoparticles of the same family for example with the results of [H Wang, C Ciret, C Cassagne, G Boudebs, “Measurement of the third order optical nonlinearities of graphene quantum dots in water at 355 nm, 532 nm and 1064 nm” Optical Materials Express 9 (2), 339-351].
On the other hand, at line 227, instead of stating that the NLO signal of ethanol solvent at 532 nm was very small, we can give the n2 value of ethanol in the same experimental conditions as used in this paper (ps regime) from reference [H Wang, C Ciret, JL Godet, C Cassagne, G Boudebs, "Measurement of the optical nonlinearities of water, ethanol and tetrahydrofuran (THF) at 355 nm" Applied Physics B 124 (6), 1-6]. Even if it is not the same wavelength, we are in the transparency region and the value will remain approximately the same.
After equation 3 on line 273 the parameter “S” is not defined in the text!
Besides it would be preferable to put everything in international unit the “esu” is a unit of another age and once we have n2 and beta (in m2/W and m/W respectively) there is no need to calculate the imaginary part of qui3 and its real one as well as qui3, it is a redundancy of information. Set something meaningful in these columns such as alpha and I0.
This is an article that deserves to be published provided that the suggestions made here are done.
Reviewer 2 Report
I have gone through the article and found that it needs major mandatory revision before acceptance. My comments are as follows:
Abstract needs to be rewritten with more specific information
1- Introduction needs to be modified with more previous reports citations such as: Introduction missing the motivation of study and also needs to be mentioned about the previous work on title material if any if not mention that too.
2-XRD figure is not justified with the work it looks amorphous not crystalline and how the phase was confirmed ??? which peak is related to rGO not shown ??
3- Raman spectra for all samples looks same, author should include Ra,man spectra for pure Sb2S3 and then compare with other, there is no confirmation related to composite formation???
4- In optical study for detemining the energy gap some missing references should be included related to used Tauc's equation:
5- The nonlinear properties should be compared properly with the exixting literature please include a table for better and visible comparison
6- Conclusion is not attractive
Overall needs major mandatory revision and re-inspection of the article
Reviewer 3 Report
The manuscript is written well, there are some revisions needed before publication. It can be accepted after addressing the below comments.
- Section 3.3. "Nonlinear Optical Proper" Heading should be appropriate it should be named as "Nonlinear Optical Properties"
- In the sentence "RGO) calculated according to the above formula. From Sb 2 S 3 to 280 Sb 2 S 3 /RGO, the imaginary part of magnetic susceptibility Im?(3)changes from negative to 281 positive and the Im?(3)values of G1" the term
" the imaginary part of magnetic susceptibility" should be replaced with "the imaginary part of third-order nonlinear optical susceptibility"
3. Authors should mention the optical path length (cuvette) since the authors have carried out Z-scan.
4. What is the rayleigh range in the current Z-scan setup, this is can be mentioned in the manuscript.
5. Whether the thin sample approximation is satisfied, needs to be explained in the revised manuscript.
6. Though the samples exhibited self-focusing effect which is a sign of positive nonlinearity, How did the authors take care of the thermal effects arising from the laser, Did the authors perform the experiment in single-shot mode to avoid such effects? clarify
Round 2
Reviewer 1 Report
Unfortunately, the authors did not modify the article properly as I asked them to do. The answers to the requested changes bypass the problem and do not correct the inadequate points to be modified to ensure a paper that deserves to be.
- I am dismayed to see an absorption in the inverse of cm-1 (line 225: units of L/g.cm-1) which gives the absorption in units of "cm" when it should be in "cm-1". I do not understand.
- It does not matter that figure 6 remains in u.a., although it would have been preferable to put a unit to the absorption because it is a parameter that would influence the results. The answers to my recommendations for what remain to be changed has been neglected. Furthermore, there are additional errors in the answers: it is inadmissible to put so many digits after the decimal point to measure the intensity (line 285), when we get the order of magnitude of the intensity in GW/cm2 we are happy. Writing 1.6941 GW/cm2 shows that the authors have no idea what the uncertainty of the measurement can be, especially in NL optics!
- Moreover, to claim that in a liquid the impact points regenerate with the Brownian motion is obvious, but the authors seem to forget that the deterioration of the specimen could result from one single (sufficiently intense) laser shot. This is what could result in figure 7 a&b with an appearance of saturable absorption when it could be only the destruction of nanoparticles resulting in a higher transmission with an intensity increasingly higher as they approach the focal plane.
- Concerning the references that I asked to add, they seem to me important to read in order to extract complementary and essential information to improve this paper. References that I suggested are essential with which the paper could gain in significance if the authors consider the scientific approach inside:
- it is essential to give the result of the NL coefficients for each nanoparticle in the suspension (as it has been done in “Wang et al., “Measurement of the third order optical nonlinearities of graphene quantum dots in water at 355 nm, 532 nm and 1064 nm” Optical Materials Express 9 (2), 339-351])
- to take into account the cell’s wall and the n2 of the solvent [H Wang, et al., "Measurement of the optical nonlinearities of water, ethanol and tetrahydrofuran (THF) at 355 nm" Applied Physics B 124 (6), 1-6] without that, errors can be prejudicial to the final results of the measurement.
I choose to give authors a second chance: there has been work done in this area and for sure following these directions is recommended and advised to improve the results and so the quality of the manuscript.
Author Response
Please see the attachment.Thank you for your valuable suggestions.

Reviewer 3 Report
The authors have revised the manuscript accordingly. The paper can be now accepted for publication.
Author Response
Thank you for your valuable suggestions on our article.
Round 3
Reviewer 1 Report
The authors have modified the text appropriately but at the risk of sounding tedious, there is still one detail to be settled. I still don't understand the unit that the authors use for absorption line 225. The definition of alpha that I understand is as follows: T=exp(-alpha x L), with T, the transmission and L, the thickness of the sample. We see in this expression that alpha is in cm-1. So I think we should give the definition of alpha used in the text and expressed in L.g-1.cm-1.
Author Response
Pleaase see the attachment.
